

# Radiocesium concentrations in wild mushrooms collected in Kawauchi Village after the accident at the Fukushima Daiichi Nuclear Power Plant

Kanami Nakashima[1], Makiko Orita[1], Naoko Fukuda[2], Yasuyuki Taira[3], Naomi Hayashida[4], Naoki Matsuda[5] and Noboru Takamura[1]

[1] Department of Global Health, Medicine and Welfare, Atomic Bomb Disease Institute, Nagasaki Univerity Graduate School of Biomedical Sciences, Nagasaki, Japan
[2] Department of Radioisotope Medicine, Atomic Bomb Disease Institute, Nagasaki University, Nagasaki, Japan
[3] Nagasaki Prefecture Office, Pharmaceutical Affairs, Nagasaki, Japan
[4] Division of Strategic Collaborative Research Center for Promotion of Collaborative Research on Radiation and Environment Health Effects, Nagasaki University Graduate School of Biomedical Sciences, Nagasaki, Japan
[5] Department of Radiation Biology and Protection, Atomic Bomb Disease Institute, Nagasaki University, Nagasaki, Japan

Corresponding author
Noboru Takamura,
takamura@nagasaki-u.ac.jp

## ABSTRACT

It is well known from the experience after the 1986 accident at the Chernobyl Nuclear Power Plant that radiocesium tends to concentrate in wild mushrooms. In this study, we collected wild mushrooms from the Kawauchi Village of Fukushima Prefecture, located within 30 km of the Fukushima Daiichi Nuclear Power Plant, and evaluated their radiocesium concentrations to estimate the risk of internal radiation exposure in local residents. We found that radioactive cesium exceeding 100 Bq/kg was detected in 125 of 154 mushrooms (81.2%). We calculated committed effective doses based on 6,278 g per year (age > 20 years, 17.2 g/day), the average intake of Japanese citizens, ranging from doses of 0.11–1.60 mSv, respectively. Although committed effective doses are limited even if residents eat contaminated foods several times, we believe that comprehensive risk-communication based on the results of the radiocesium measurements of food, water, and soil is necessary for the recovery of Fukushima after this nuclear disaster.

## INTRODUCTION

On March 11, 2011, a 9.0-magnitude earthquake struck the east coast of Iwate, Miyagi, and Fukushima Prefectures in Japan. The earthquake, in combination with the resulting tsunami, triggered a severe nuclear accident at the Fukushima Daiichi Nuclear Power Plant (FNPP) (*IAEA, 2015*). Due to this accident, large amounts of radionuclides, including iodine-131 ($^{131}$I), cesium-134 ($^{134}$Cs), and cesium-137 ($^{137}$Cs) were released into the atmosphere (*UNSCEAR, 2013*). The United Nations Scientific Committee on the Effects of

Atomic Radiation (UNSCEAR) estimated the total release of $^{131}$I, $^{134}$Cs, and $^{137}$Cs at 120.0, 9.0, and 8.8 petabecquerel (PBq), respectively.

To minimize the internal radiation exposure of local residents, Japanese national and prefectural governments began to monitor selected foodstuffs (milk, vegetables, grains, meat, fish, etc.) on March 16, 2011. Those containing radioactive material that exceeded the provisional regulation values (set in the initial phase of the accident at FNPP based on the assumption that the radiation level will drop in accordance with the half-lives of radioactive materials and that the continuous intake of food contaminated with radiation at this level will not affect health) recommended on March 17, 2011, by Japan's Ministry of Health, Labour and Welfare (MHLW) were prohibited from distribution on March 21, 2011, and from consumption on March 23, 2011 (*Hamada, Ogino & Fujimichi, 2012*; *Merz, Steinhauser & Hamada, 2013*; *Merz, Shozugawa & Steinhauser, 2015*). Due to this stringent food-control policy, internal radiation from radioactive cesium and radioactive iodine was limited (*Harada et al., 2012*; *Koizumi et al., 2012*; *Hayano et al., 2013*). Fukushima Prefecture reported the results of internal radiation doses measured with a whole-body counter in residents of Fukushima Prefecture and its evacuees from June 2011 to February 2015. Of these, 233,199 (99.9%) showed a committed effective dose of <1 mSv; the maximum recorded level was 3 mSv, which was measured in two individuals (*Taira et al., 2014*).

However, based on the experience after the 1986 accident at the Chernobyl Nuclear Power Plant, it is well known that radiocesium tends to concentrate in wild mushrooms (*Skuterud et al., 1997*; *Jesko et al., 2000*; *Hille et al., 2000*; *Hoshi et al., 2000*; *Travnikova et al., 2001*; *Sekitani et al., 2010*; *Guillén & Baeza, 2014*). *Hoshi et al. (2000)* reported that children consuming mushrooms showed a high $^{137}$Cs body burden near Chernobyl, which suggests that mushrooms are one of the main contributors to internal radiation exposure from radiocesium after nuclear disasters. Although the evaluation of radiocesium concentrations in wild mushrooms is important, comprehensive studies have not been conducted to measure the concentrations of radiocesium in mushrooms in a certain area and to clarify the risk of internal exposure in the residents of Fukushima. In this study, we collected wild mushrooms from the Kawauchi Village of Fukushima Prefecture, located within 30 km of the FNPP (Fig. 1), and evaluated their radiocesium concentrations to estimate the internal radiation exposure of local residents.

## MATERIALS & METHODS

### Sampling of mushrooms

FNPP is located on the east coast of Honshu Island, approximately 200 km northeast of Tokyo, Japan. Samples around the FNPP were collected at Kawauchi Village (Village Office, N37°20′, E140°48′).

Prior to the study, we obtained approval from the Kawauchi Village Office (Approval Number: 20130120). From September to November 2013, we asked residents of the village to collect mushrooms and show the location of collection of each mushroom. 154 mushroom samples from a total of 22 species were collected in the village.

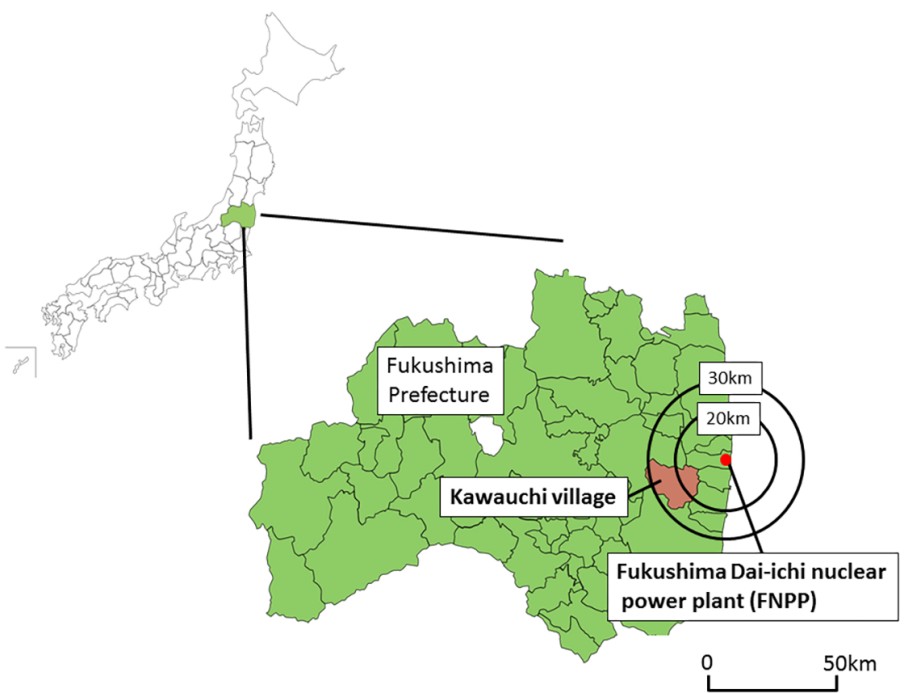

**Figure 1** Location of Kawauchi Village, Fukushima Prefecture.

After collection, all samples were washed by water to remove soils and dried in a fixed-temperature dryer (60 °C for 24 h, then 105 °C for 1 h). They were then placed in plastic containers made of polypropylene and analyzed with a high-purity germanium detector (ORTEC®, GMX30–70, Ortec International Inc., Oak Ridge, TN, USA) coupled with a multi-channel analyzer (MCA7600, Seiko EG&G Co., Ltd., Chiba, Japan) for 3,600 s. The measuring time was set to detect the objective radionuclide, and the gamma-ray peaks used for the measurements were 604.66 keV for $^{134}$Cs (2.1 y) and 661.64 keV for $^{137}$Cs (30 y). Decay corrections were made based on the sampling date, and detector efficiency calibration was performed for different measurement geometries using mixed-activity standard volume sources (Japan Radioisotope Association, Tokyo, Japan). Concentrations of radiocesium were automatically adjusted by the date of collection, and the data was defined as the concentration at the collection date. All measurements were performed at Nagasaki University (Nagasaki, Japan). The sum of $^{134}$Cs and $^{137}$Cs concentrations was indicated as "concentrations of radiocesium." In most samples, mushrooms contained $^{134}$Cs and $^{137}$Cs in a fixed ratio. However, in some samples only $^{137}$Cs samples were reported, since as $^{134}$Cs concentrations were below the detection limit. For such samples, concentrations of $^{137}$Cs were indicated as "concentrations of radiocesium." We evaluated the ratio of radiocesium concentrations in dried and raw mushrooms using 81 mushrooms, finding that it correlates strongly ($r = 0.913$, $p < 0.001$, Fig. 2) and obtaining the following formula:

Radiocesium concentrations in raw mushroom (Bq/kg)

= 0.136*{radiocesium concentrations in dried mushroom (Bq/kg)}

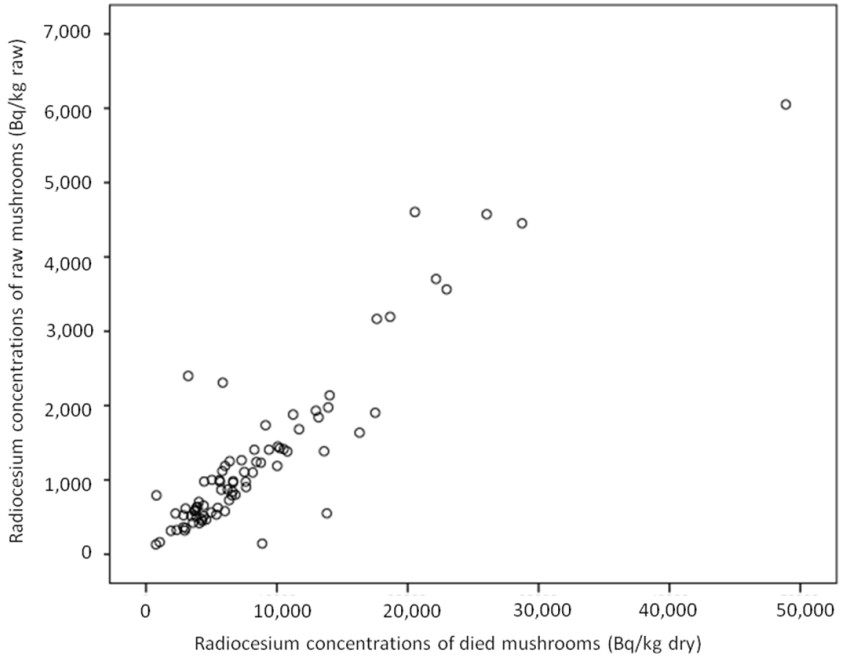

**Figure 2** Relationship of radiocesium concentrations between dried and raw mushrooms.

In this manuscript, we indicated the concentrations of raw mushrooms and used them for analysis.

## Committed effective dose

The committed effective dose from mushroom concentration intake was calculated using the following formula:

$$H\text{int} = C \cdot ED\text{int} \cdot e$$

where $C$ is the activity concentration (median) of detected artificial radionuclides (radio-cesium) (Bq/kg), $D\text{int}$ is the dose conversion coefficient for adult intake (age $>$ 20 years, $1.9 \times 10^{-5}$ mSv/Bq for $^{134}$Cs and $1.3 \times 10^{-5}$ mSv/Bq for $^{137}$Cs) (*ICRP, 1995*), and $e$ is the daily intake value (age $>$ 20 years, 17.2 g/day, the average intake of Japanese citizens) (*MHLW, 2012*). We assumed that annual intake could be attributed to each species.

## RESULTS

The species and amounts of the collected wild mushrooms are listed in Table 1. Among the 154 mushroom samples from 22 species, 79 (51.3%) were *Sarcodon aspratus*, 11 (7.1%) were *Hypholoma sublateritium*, and 10 (6.5%) were *Armillaria mellea*. The concentrations of radiocesium ($^{134}$Cs $+$ $^{137}$Cs) in mushrooms are shown in Fig. 3. Radiocesium was not detected in 19 mushrooms (12.3%); however, 100–999 Bq/kg of radiocesium was detected in 69 mushrooms (44.8%), and $>$1,000 Bq/kg of radiocesium was detected in 56 mushrooms (36.4%). The maximum concentration was 5,432.7 Bq/kg of $^{134}$Cs and 11,616.2 Bq/kg of $^{137}$Cs in *Cortinarius salor Fr.*

**Table 1** Concentrations of radiocesium in wild mushrooms.

| Species (habitat) | $n$ | $^{134}$Cs[*] | $^{134}$Cs (Bq/kg)[**] | $^{137}$Cs[*] | $^{137}$Cs (Bq/kg)[**] |
|---|---|---|---|---|---|
| *Sarcodon aspratus* (surface soil) | 79 | 79 | 290.8 (43.8–1,870.1) | 79 | 667.5 (126.8–4,504.5) |
| *Hypholoma sublateritium* (wood) | 11 | 9 | 177.5 (10.3–509.2) | 10 | 388.3 (22.4–1,087.9) |
| *Armillaria mellea* (wood) | 10 | 6 | 46.3 (31.1–189.7) | 8 | 69.8 (16.9–394.5) |
| *Tricholoma matsutake* (surface soil) | 6 | 6 | 139.8 (70.0–491.4) | 6 | 299.6 (177.5–1,171.2) |
| *Pholiota microspore* (wood) | 6 | 6 | 298.9 (74.5–1,706.3) | 6 | 652.5 (185.6–3,685.5) |
| *Lyophyllum shimeji* (surface soil) | 5 | 3 | 89.8 (64.8–107.0) | 4 | 164.5 (95.7–193.8) |
| *Lyophyllum decastes* (surface soil) | 5 | 1 | 18.4 | 2 | 16.7 (15.4–17.9) |
| *Cortinarius salor Fr.* (surface soil) | 4 | 4 | 3,596.8 (363.1–5,432.7) | 4 | 7,589.4(802.6–11,616.2) |
| *Boletopsis leucomelas* (surface soil) | 4 | 4 | 191.3 (69.7–763.0) | 4 | 444.3 (180.2–1,760.9) |
| *Pholiota squarrosa* (wood) | 4 | 0 | ND | 1 | 56.0 |
| *Hygrophorus russula* (surface soil) | 3 | 3 | 415.0 (319.4–2,661.5) | 3 | 986.9 (727.6–5,719.4) |
| *Grifola frondosa* (wood) | 3 | 0 | ND | 0 | ND |
| *Phaeolepiota aurea* (surface soil) | 3 | 0 | ND | 0 | ND |
| *Suillus bovinus* (surface soil) | 2 | 2 | 631.3 (587.0–675.7) | 2 | 1,352.7(1,272.2–1,433.3) |
| *Lyophyllum fumosum* (surface soil) | 2 | 2 | 84.4 (68.1–100.6) | 2 | 153.4 (93.8–212.9) |
| *Lepista sordida* (surface soil) | 1 | 1 | 4,927.7 | 1 | 10,415.0 |
| *Lepista nuda* (surface soil) | 1 | 1 | 2,975.7 | 1 | 6,429.2 |
| *Panellus serotinus* (wood) | 1 | 1 | 35.5 | 1 | 88.7 |
| *Pleurotus ostreatus* (wood) | 1 | 1 | 23.5 | 1 | 57.2 |
| *Lentinula edodes* (wood) | 1 | 0 | ND | 0 | ND |
| *Armillaria tabescens* (wood) | 1 | 0 | ND | 0 | ND |
| *Entoloma sarcopum* (surface soil) | 1 | 0 | ND | 0 | ND |

**Notes.**
[*] Number of detected mushrooms.
[**] Median (minimum–maximum).
ND, Not detected.

The concentration of radiocesium of each species is shown in Table 1. Radiocesium was detected in 79 of 79 samples (100%) of *Sarcodon aspratus*, 10 of 11 samples (90.9%) of *Hypholoma sublateritium*, 8 of 10 samples (80%) of *Armillaria mellea*, 6 of 6 samples (100%) of *Tricholoma matsutake*, and 6 of 6 samples (100%) of *Pholiota microspora*. However, radiocesium was not detectable in *Grifola frondosa* ($N = 3$), *Lentinula edodes* ($N = 1$), *Armillaria tabescens* ($N = 1$), and *Entoloma sarcopum* ($N = 1$). All 79 samples exceeded 100 Bq/kg in *Sarcodon aspratus* (Fig. 4), whereas 46 of 75 samples exceeded (61.3%) 100 Bq/kg in other mushrooms (Fig. 5).

Next, we mapped the distribution of mushrooms with concentrations of radiocesium in Kawauchi Village (Fig. 6). The calculated committed effective doses are shown in Table 2. Among 125 mushrooms that contained radiocesium exceeding 100 Bq/kg, the minimum and maximum calculated annual committed effective doses ranged from 0.11–1.60 mSv/year.

## DISCUSSION

In 2013, MHLW reported that mushrooms in Fukushima Prefecture contained low concentrations of radiocesium (less than 60 Bq/kg) in Fukushima (*MHLW, 2013*), but

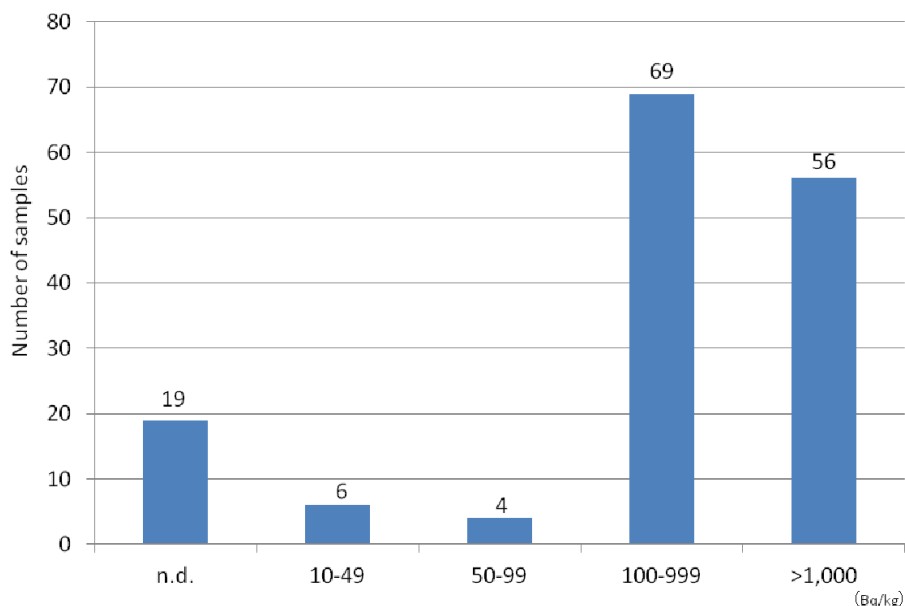

**Figure 3** Distribution of radiocesium concentrations ($^{134}Cs + ^{137}Cs$) in all mushrooms.

**Table 2** Committed effective dose due to wild mushroom intake.

| Species | n | mSv/year[*] |
|---|---|---|
| *Sarcodon aspratus* | 79 | 0.09 (0.02–0.59) |
| *Hypholoma sublateritium* | 10 | 0.05 (<0.01–0.15) |
| *Armillaria mellea* | 8 | 0.01 (<0.01–0.05) |
| *Tricholoma matsutake* | 6 | 0.04 (0.02–0.15) |
| *Pholiota microspora* | 6 | 0.09 (0.02–0.50) |
| *Lyophyllum shimeji* | 4 | 0.03 (0.01–0.03) |
| *Lyophyllum decastes* | 2 | <0.01 (<0.01–<0.01) |
| *Cortinarius salor Fr.* | 4 | 1.05 (0.11–1.60) |
| *Boletopsis leucomelas* | 4 | 0.06 (0.02–0.23) |
| *Pholiota squarrosa* | 1 | <0.01 |
| *Hygrophorus russula* | 3 | 0.13 (0.10–0.78) |
| *Suillus bovinus* | 2 | 0.19 (0.17–0.20) |
| *Lyophyllum fumosum* | 2 | 0.03 (0.02–0.03) |
| *Lepista sordida* | 1 | 1.44 |
| *Lepista nuda* | 1 | 0.88 |
| *Panellus serotinus* | 1 | 0.01 |
| *Pleurotus ostreatus* | 1 | 0.01 |

**Notes.**
[*] Median (minimum–maximum).

did not report on wild mushrooms in this report. Recently, the International Atomic Energy Agency (*IAEA, 2015*) published technical reports on the accident at FDNPP, and pointed out that the concentration of radiocesium in wild mushrooms was higher than in agricultural products in Fukushima.

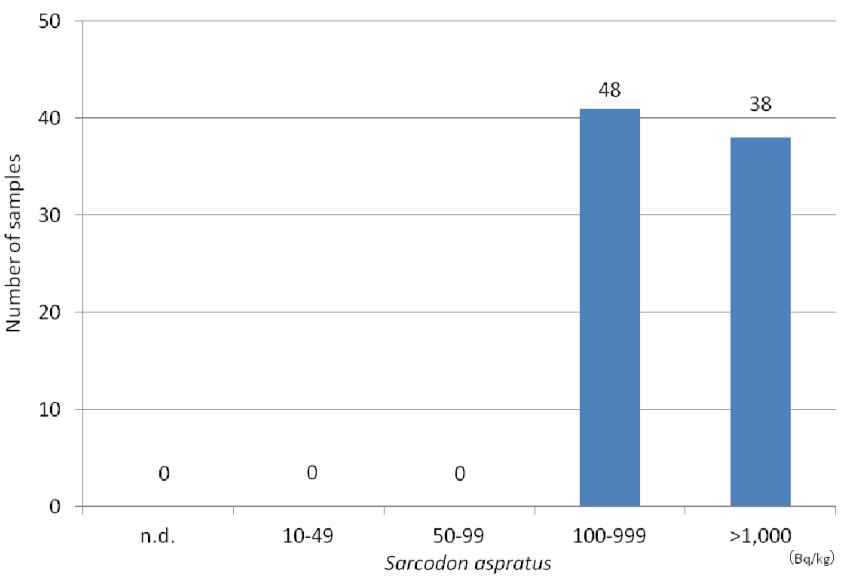

**Figure 4** Distribution of radiocesium concentrations ($^{134}$Cs+$^{137}$Cs) in *Sarcodon aspratus.*

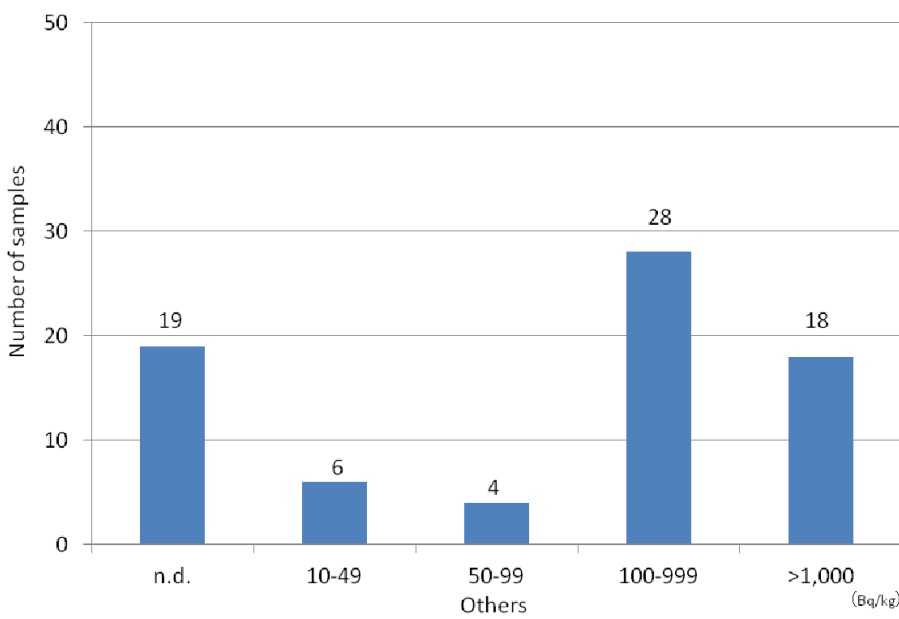

**Figure 5** Distribution of radiocesium concentrations ($^{134}$Cs+$^{137}$Cs) in other mushrooms.

In this study, radioactive cesium exceeding 100 Bq/kg (the current regulated value of radiocesium for mushroom) was detected in 125 of 154 (81.2%) mushrooms collected in Kawauchi Village, Fukushima Prefecture. Kawauchi Village is located 20–30 km southwest of the FNPP, and most of its residents were evacuated during the initial phase of the accident at the FNPP. On January 31, 2012, the head of Kawauchi Village declared that residents who lived at least 20 km away from the FNPP could return to their homes, after the Japanese Prime Minister declared that the reactors had achieved a state of "cold shutdown" in December 2011 (*Orita et al., 2013*). Since this declaration, the village office

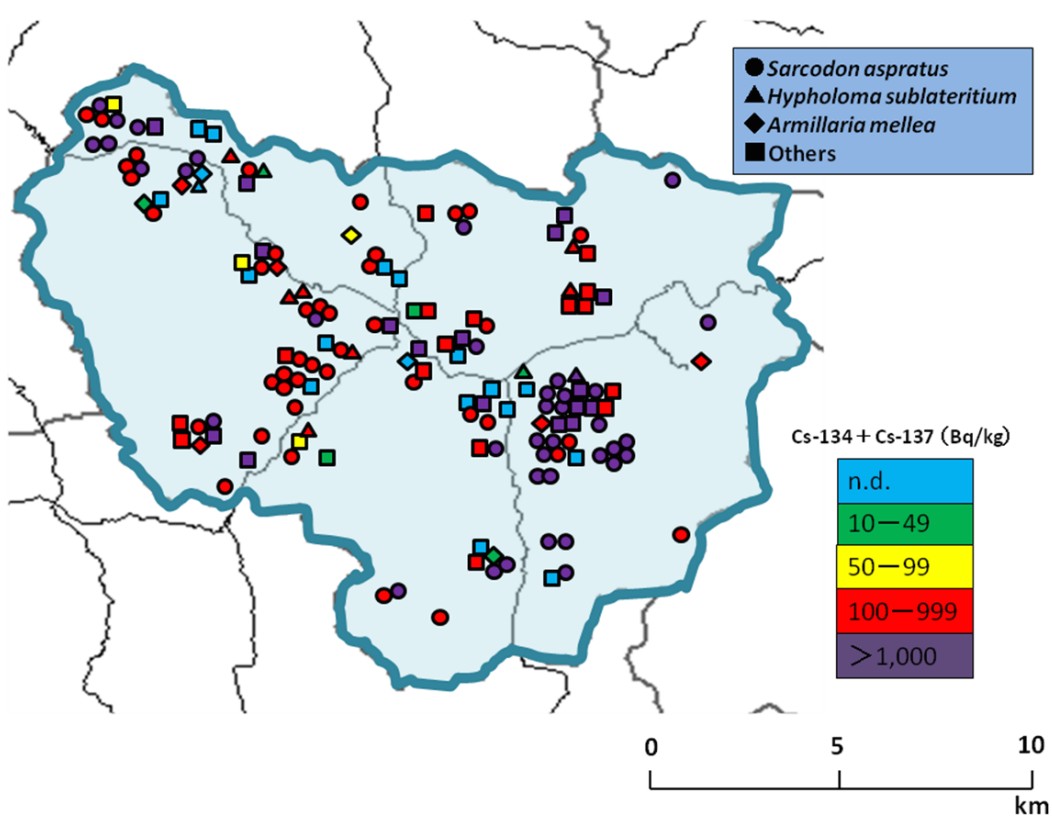

**Figure 6** Map of mushroom concentrations in Kawauchi Village.

has been working steadily towards reconstruction, including decontamination within the village. Decontamination of residential houses has been conducted since mid-2011, even within the 20 km radius of the plant. However, decontamination has not been conducted in the forests of Fukushima Prefecture, including Kawauchi Village. Therefore, because all of the mushrooms in this study were collected in the forest around the village, high frequencies of radiocesium were detected.

Since the Chernobyl accident, a series of studies has been conducted to clarify the influence of radiocesium in forest-derived products, including mushrooms. *Kaduka et al. (2006)* analyzed the $^{137}$Cs aggregated transfer factor from the soil to different mushrooms and showed that the aggregated transfer factors depend on the mushroom's trophic group, biological family, genus, and species (*Bannai, Yoshida & Muramatsu, 1994*). *Bulko et al. (2014)* evaluated the $^{137}$Cs uptake by forest-derived products in the Gomel region, which was the most heavily contaminated after the Chernobyl accident, and found that the accumulation of $^{137}$Cs in mushrooms and berries was directly related to the radiocesium contamination density of the soil, which is accounted for both by the form of the Chernobyl fallout and by the natural and climatic conditions that determine variations in the availability of radionuclides in the soil.

Although the number of samples was limited, we found that 17 species showed higher concentrations of radiocesium and that five species showed lower concentrations. This suggests that the concentration of radiocesium might depend on the species of

mushroom. Usually, habitat varies depending on species. To determine the factors affecting radiocesium concentrations in mushrooms, *Yoshida & Muramatsu (1994)* collected mushrooms and categorized them into four different groups according to the main habitat of their mycelia: wood, the litter layer, the surface soil layer (0–5 cm), and the following soil layer (>5 cm), and found that the surface soil layer group showed the highest average concentrations of $^{137}$Cs. They concluded that the habitat of the mycelium seemed to be one of the most important factors controlling radiocesium concentration in mushrooms. On the other hand, although the sample size was limited in our study, there was no clear relationship between habitat and radiocesium concentration in mushrooms. Further follow-up studies are required to clarify the determinants of radiocesium concentrations in mushrooms.

We calculated committed effective doses ranging from 0.11–1.60 mSv, based on the average annual intake of mushrooms in Japanese citizens. After the FNPP accident, the Japanese government established provisional regulation values for $^{131}$I (300 Bq/kg for drinking water and milk, 2,000 Bq/kg for vegetables) and for $^{134}$Cs and $^{137}$Cs (200 Bq/kg for drinking water and milk, 500 Bq/kg for vegetables, grains, meat, fish, and eggs). Due to this strict food-control policy, internal exposure from radioactive cesium and radioactive iodine were relatively limited (*Harada et al., 2012*; *Koizumi et al., 2012*; *Hayano et al., 2013*). However, according to the experience after the Chernobyl accident, excess intake of contaminated mushrooms is a risk factor for internal radiation exposure among residents. *Hoshi et al. (2000)* conducted measurements of the $^{137}$Cs body burden in 1991–1996 for children residing in Bryansk Oblast (Russian Federation), an area that experienced contamination following the Chernobyl accident, and discovered that the most common food items contributing to $^{137}$Cs intake in children were mushrooms, meat, milk, and vegetables. *Travnikova et al. (2001)* also found that the individual content of $^{137}$Cs in the bodies of inhabitants of a village in Bryansk Oblast correlated with their consumption of milk during the initial period after the accident and with their consumption of forest mushrooms during the subsequent period. Although our results showed that committed effective doses were limited even if residents ate contaminated foods several times, a long-term risk evaluation for the internal radiation exposure of these individuals is needed in order to gain a better understanding of radiation safety in Fukushima.

There are several limitations of this study. First, it was conducted only in Kawauchi Village for one year. Second, we could not evaluate the relationship between radiocesium concentrations in mushrooms and soil, due to insufficient soil samples. Further comprehensive studies are necessary to evaluate the concentrations of radiocesium in mushrooms in Fukushima after the FNPP accident.

## CONCLUSIONS

We evaluated radiocesium concentrations in wild mushrooms collected at Kawauchi Village in Fukushima Prefecture and found that radiocesium exists in wild mushrooms

at a high frequency. Although committed effective doses are limited even if residents eat contaminated foods several times, we believe that comprehensive risk-communication based on measurements of radiocesium in the food, water and soil, is necessary for the recovery of Fukushima after the nuclear disaster.

### Funding

This study was supported by the research projects on health effects of nuclear disasters of the Ministry of the Environment and by a research grant (No. 15K00540) from the Japan Society for the Promotion of Science (JSPS). The funders had no role in study design, data collection and analysis, decision to publish, or preparation of the manuscript.

### Grant Disclosures

The following grant information was disclosed by the authors:
Ministry of the Environment.
Japan Society for the Promotion of Science (JSPS): 15K00540.

### Competing Interests

The authors declare there are no competing interests.

### Author Contributions

- Kanami Nakashima conceived and designed the experiments, performed the experiments, analyzed the data, wrote the paper, prepared figures and/or tables.
- Makiko Orita performed the experiments, analyzed the data.
- Naoko Fukuda performed the experiments, contributed reagents/materials/analysis tools.
- Yasuyuki Taira conceived and designed the experiments.
- Naomi Hayashida and Naoki Matsuda conceived and designed the experiments, contributed reagents/materials/analysis tools, reviewed drafts of the paper.
- Noboru Takamura analyzed the data, contributed reagents/materials/analysis tools, wrote the paper, prepared figures and/or tables, reviewed drafts of the paper.

### Field Study Permissions

The following information was supplied relating to field study approvals (i.e., approving body and any reference numbers):

1. Kawauchi Village Office, Fukushima, Japan

2. Prior to the study, we obtained approval to conduct current study from Kawauchi Village Office (Approval Number: 20130120).

### Data Availability

All raw data is provided in Supplemental Information 1.

## Supplemental Information

Supplemental information for this article can be found online at http://dx.doi.org/10.7717/peerj.1427#supplemental-information.

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
