# Peer review of "Radiocesium concentrations in wild mushrooms collected in Kawauchi Village after the accident at the Fukushima Daiichi Nuclear Power Plant"

_PeerJ, doi:10.7717/peerj.1427_

## Round 0.1 · original submission · Major Revisions

· Academic Editor

Major Revisions

The biggest concern raised by the reviewers is the clarity and validity of calculations of dose from mushrooms because of the confusion surrounding use of wet and dry weights, and validation against other published data. Please pay particular attention to this area.

Your revised manuscript should address all the points raised by the reviewers individually, as well as my own comments below:

The term "relatively" is used throughout the manuscript, without quantification, e.g. lines 30, 53, 63, 118, 151, 166, 176, 187, 188). Please clarify what is meant. For instance you claim that five species had "relatively lower concentrations of radiocesium" - does this refer to species in which a level of ND was reported? If not, what are your criteria for "relatively higher" and "relatively lower"? Do the mushrooms fall into two clusters or is the distinction arbitrary? What is the accuracy, resolution and limit of detection for your method. What quality control measures were employed? Were samples measured more than once?

Line 49 What were the provisional regulation values?

Line 79 Were the mushrooms washed or was there carryover of soil, etc?

Line 85 Please detail the corrections based on sampling date. How were these done. Were they corrected from measurement date to sampling date? Sampling date to date of the Fukushima accident?

Line 89 What is the significance, if any, of some mushrooms (were these individuals or species?) containing only Cs-137? Are the isotopes typically found in a fixed ratio? If so, was this a function of the detection limit or do you think there is a metabolic significance?

Line 100 As noted by the reviewers, please state unambiguously when you are referring to wet and dry weights.

Line 106 Since these samples were taken 30-32 months post-accident, are there any significant daughter radionucleides that should be considered? How does this work build upon your group's previous publication (Taira et al. 2014)?

Line 121 What is the source of the figures for mushroom consumption in the Japanese population? Is this nationwide? Is the figure applicable to residents of the village studied? Is the consumption seasonal or spread through the year? How might this be affected by how radiocesium is handled in humans - i.e. does it accumulate or is it excreted?

Line 128 What is the significance of the 100 Bq/kg?

Line 160 You state that the habitat in which mushrooms grow is correlated with radiocesium content, but you do not indicate or discuss the habitats of the species you examined (see comments from Reviewer 1)

Table 1 How were the concentrations distributed? Is the mean the most appropriate metric, or would median be more appropriate? It might be useful to differentiate the concentrations of Cs-134 and Cs-137 in the table, as well as indicate habitats.

Table 2 It seems redundant to express doses in both uSv/day and mSv/year. Whis is more representative, bearing in mind my earlier comments about possible seasonal distribution of mushroom intake and the residence time of radiocesium in the human body? What are the assumptions made in calculating the doses? For instance, do you assume that the entire average daily mushroom intake is attributed to each species separately?

Figure 1 How was the sampling site selected? How does it relate to the site of the accident in terms of prevailing winds, etc.?

Figure 2 There are no error bars to indicate the range of the data. It would be useful if the data were broken down on the basis of species, or of habitat.

Figure 1 and 3 should include scale bars to indicate distances.

Figure 2 Sampling appears to be concentrated along water courses. Does this reflect the habitats where the mushrooms grow, or is it a function of sampling effort?

·

Basic reporting

This is an interesting manuscript that is written in clear, concise English. The manuscript reports on radioactive cesium concentrations in mushrooms growing within areas contaminated from the Fukushima Power Plant disasters in Japan.
The authors do a nice job of introducing the importance of mushroom consumption by humans as a known pathway for enhanced radioactivity ingestion, and document previous findings from the Chernobyl accident in the Ukraine. Although not stated by the authors, mushroom consumption is also thought to be a major contributor to the contamination of several wildlife species, particularly deer and wild boar. Thus some readers of the journal will be interested in the results of this paper from that perspective as well.

Experimental design

I recommend that the paper be accepted for publication following some minor clarifications that will avoid potential confusion among some readers.
Specifically: (all line referrals are from the pdf version of the manuscript)
• Additional information is needed on dry versus fresh mass of the mushrooms. It is not always clear when dry versus fresh mass were used.
o On lines 30, and again on line 100, please state if the ingestion rates for mushrooms (grams / day) are based on fresh or dried mass basis.
o On line 28, and again on line 128, the authors state that the 100 Bq/kg consumption limit set by the Japanese government was exceeded 87% of the time, based on dried mushroom activity concentrations. However, I suspect that many of these mushrooms are prepared fresh and would thus have a significantly lower activity concentration. The authors should clarify, and preferably report their data in a format (fresh vs dried) that is most appropriate for the mushroom consumption practices by humans in Japan. Adding a couple of lines of text on this matter would be helpful.
o Lines 95-100 need clarification on how the authors converted dried to wet mass when calculating dose. Some information is presented further down in the manuscript (i.e. line 131) but the manner in which dried and wet mass are dealt with should be presented concisely within the methodology section.
o All of the above could be easily cleared up by reporting activity concentrations more clearly. For example rather than stating Bq / kg, state Bq / kg dry mass, or Bq / kg wet mass. And by clearly stating dry or wet mass ingestion rates (i.e. grams per day wet or grams per day dry).
• Consider enhancing Table 1 by adding a column that indicates the main habitat of the mycelia (i.e. wood, litter layer, surface soil, or > 5 cm in soil) as is discussed on line 169 of the Discussion section.
• The Committed effective doses are presented with too many significant figures, considering rounding to two significant figures.
• Lines 162 – 167 present interesting information on dietary limits set by the Japanese government; however, the current 100 Bq/kg limit is not included. Please extend this discussion to explain the current limit.

Validity of the findings

The organization, structure and methodology of the work are all well done. The results are supported by the information given, and the figures and tables are not excessive, but add to the manuscript.

·

Basic reporting

The current report concerning with study of "Radiocesium concentrations in wild mushrooms collected in Kawauchi Village after the accident at the Fukushima Daiichi Nuclear Power Plant. I see that the report is explaining the effects of radiation in biological systems like mushroom. This is an important topics because it gives an idea about the limit of radionuclide accumulation inside edible biological systems with radiation accidents like Fukushima Daiichi Nuclear Power Plant explosion.

Experimental design

The design of report experiments is well carried out. But, I see that this study will be much more better if it is extended for two or three years to get more information about the decay of that radionuclide.

Validity of the findings

The duration of experiments should extend for more than two years. Also,soil is not only the source of radionuclide contamination. There is water, air ......etc

Additional comments

1-Soil is not the only the source of radionuclide contamination.
2-One year is not enough to get a good impression about the nature of radionuclide decay.
3-Why there is no analysis for the most heavily and durable mushroom species for its secondary metabolites like different pigments to explain why there is species with high radionuclide contents ?!!!!!

Reviewer 3 ·

Basic reporting

see comments in general comments

Experimental design

see comments in general comments

Validity of the findings

see comments in general comments

Additional comments

The present paper aimed to measure radioactivity of various wild mushrooms in Kawauchi Village and to assess committed effective dose by intake of the mushrooms. The authors found that the 87.8 % of the samples were exceeded the regulation value in Japan, however estimated committed effective doses were ranging from 0.093-1.368 mSv/year.
However, the authors were discussed the activity concentration in mushroom with dry weight base, whereas the regulation value was derived for fresh weight based activity in foods. Additionally, daily ingestion rate of mushroom of MHLW survey which is adopted in the paper is also fresh weight, therefore estimated committed effective dose is overestimation.
In conclusion, the authors said that "..found that radiocesium exists in wild mushrooms at a relatively high frequency", but I could not found other distribution of the concentration in mushrooms. To validate the measurement, I strongly recommend that the present data should be compared with MHLW food monitoring data in 2013. Almost all of mushrooms in MHLW monitoring data are Lentinula edodes and wild mushrooms are rare, but some tendency could be found. Recently published IAEA report is also discussed about the wild mushroom using with monitoring data, so that the IAEA report also should be referred.

Other comments were listed below.
1. The all-age averaged daily ingestion rate was used whereas the dose conversion coefficient for adult was used. According to MHLW National Health and Nutrition survey 2012, the average value of the mushroom for over 20 is 17.2 g/day.
2. Line 111: Table 2 might be Table 1.
3. Line 117: Fig.3 might be Table 1.
4. Spatial distribution in Kawauchi Village had not well discussed, whereas figure 3 is included some information.

---

## Round 0.2 · Minor Revisions

· Academic Editor

Minor Revisions

Thank you for addressing the comments on your original submission.

I still have some concerns about Figure 1 - the map showing the location of Kawauchi village. (i) The circles marked as 20 km and 30 km appear to be 20 and 30 km in diameter (according to the scale bar). It is more usual to give distances in this situation as a radius from the center. Please amend the circle labels, or the scale bar - whichever is incorrect. Also (ii) the circles do not seem to be centered on the power plant (the red marker is above and to the right of the center).

In addition, please attend to the error pointed out by Reviewer 1 on lines 58-59 and check the manuscript carefully for any other such issues before resubmitting.

·

Basic reporting

this is a follow-up review following revisions by the authors....see original for this section.

Experimental design

this is a follow-up review following revisions by the authors....see original for this section.

Validity of the findings

this is a follow-up review following revisions by the authors....see original for this section.

Additional comments

This is a follow up review following revisions by the authors. The authors have addressed the original concerns. Thank you.

I did see a typographical error....line 59 of the pdf version is not a complete sentence, and the subject-verb agreement is not correct.

Reviewer 3 ·

Basic reporting

no comments

Experimental design

no comments

Validity of the findings

no comments

Additional comments

All comments I suggested were corrected appropriately so that I suggest the manuscript is satisfied acceptable to publish.

---

## Round 0.3 · accepted · Accept

· Academic Editor

Accept

Thank you for your prompt attention to the revisions.